# Controllable assembly of sub-1 nm nanowires for the construction of aerogels

Yuxiang Du[1,2], Yueyue Xiu[1,2], Xin Yang[1,2], Rui Fu[1,2], Dan Liu[1,2], Lipeng Liu[1,2] & Huazheng Sai [1,2] ✉

Aerogels exhibit excellent properties owing to their nano-sized building blocks and unique structures. With increasing application demands, traditional nanoscale building blocks have limitations in further optimizing the performance of aerogels; therefore, the development of novel, high-performance building blocks has become an urgent challenge in the field. Sub-1 nm nanowires (SNWs) exhibit polymer-like properties that make them superior to nanoscale nanowires, and are well-suited as new building blocks for aerogels. In this study, we achieved precise control over the aggregation state of GdOOH SNWs (Gd–SNWs) in three-dimensional space by regulating the interactions between SNWs as well as between SNWs and solvents, thereby obtaining SNW aerogels (SNWAs) with low density ( ~ 0.024 g cm$^{-3}$) and high specific surface area (505 m$^2$ g$^{-1}$). After silanization, the superhydrophobic SNWAs exhibited excellent fatigue resistance (50 cycles with a set strain of 50%). This innovative approach enriches the types of aerogels building blocks and opens up new avenues for high-performance aerogels.

In 2022, the International Union of Pure and Applied Chemistry (IUPAC) designated aerogels as one of the top ten emerging technologies in chemistry[1]. These materials uniquely bridge the microscopic and macroscopic scales by integrating the characteristic properties of nanoporous frameworks with those of nanoscale building blocks, including nanowires, nanosheets, nanotubes, and nanoparticles[2–6]. However, further optimization of key aerogel properties such as specific surface area, thermal conductivity, and density remains constrained by the fundamental limitations inherent in conventional nanoscale building blocks[7]. Consequently, the development of innovative high-performance building blocks is a critical challenge in advanced materials research.

Sub-1-nm nanowires (SNWs) represent a distinct class of nanowires characterized by diameters approaching single-crystal cell dimensions[8,9]. Unlike conventional nanowires, inorganic SNWs exhibit unique size effects because of their similarity to polymer molecular scales, demonstrating exceptional properties such as ultrahigh specific surface area, enhanced flexibility, and elevated surface energy[10–12]. To translate these advantages into functional macroscopic materials,

researchers have employed assembly techniques such as wet spinning and electrospinning[13]. Nevertheless, macroscopic materials constructed with SNWs are still limited in terms of variety and structural dimensions, and are mainly confined to 1D fibers and 2D films[14,15]. The polymer-like properties of SNWs render them suitable building blocks for aerogels. When SNWs are used as building blocks for aerogel materials, their sub-1-nm size properties are expected to further optimize the properties of aerogels. This innovative approach not only enriches the variety of aerogel building blocks but also affords a new approach for the transformation of SNWs into macroscopic materials with practical applications.

Although the assembly of SNWs into aerogel materials has recently been realized by freeze-casting[16], the uncontrolled aggregation and adhesion of SNWs caused by ice crystal extrusion resulted in the aerogel having a small specific surface area (only ~20 m$^2$ g$^{-1}$)[17], which does not fully reflect the potential advantages of SNWs, such as a high specific surface area. Therefore, achieving a controllable adjustment of SNWs assembly into aerogels is crucial for sub-1-nm nanowire aerogels (SNWAs) to exhibit the characteristics of sub-1-nm size

[1]School of Chemistry and Chemical Engineering, Inner Mongolia University of Science & Technology, Baotou, China. [2]Aerogel Functional Nanomaterials Laboratory, Inner Mongolia University of Science & Technology, Baotou, China. ✉e-mail: shz15@tsinghua.org.cn

materials. The preferred method is to first construct a wet gel using SNWs as building blocks, followed by supercritical drying to obtain an aerogel. To achieve this, SNWs constructed as stable wet gels are indispensable prerequisites. Although SNWs can be assembled into an organic gel via electrostatic and van der Waals forces in polar solvents[8,18], the framework of this organic gel collapses during supercritical drying and cannot form a super porous structure. This is because the ultrafine SNWs are extremely fragile in rapidly changing environments. However, polymer molecules similar in size to SNWs, such as cellulose and agarose, are capable of gelation followed by supercritical drying to produce aerogels[19,20]. This is mainly because polymers undergo a certain degree of molecular chain segment aggregation during the gelation process, resulting in a tougher gel skeleton than that existing between dispersed individual molecular chains. This phenomenon indicates that achieving the controllable assembly of SNWs to obtain a certain strength is an effective strategy to overcome the insufficient mechanical strength of the gel skeleton.

We herein propose a strategy for the controlled aggregation and assembly of GdOOH SNWs (Gd–SNWs) in polar solvents to fabricate metal oxide aerogels. The ligands on the surface of SNWs were adjusted to achieve the uniform dispersion of the SNWs in polar solvents. In addition, the solvent polarity can control the aggregation state of the SNWs to a certain extent. The introduction of organic acids enabled the assembly of the SNWs into a 3D interconnected network, thereby successfully constructing GdOOH SNWAs (Gd–SNWAs). Gd–SNWA synthesized via supercritical drying exhibits integrated properties of ultralow density (~0.024 g cm$^{-3}$), high specific surface area (505 m$^2$ g$^{-1}$), superior light transmittance (Fig. 1a), excellent strain-recoverable compressibility, and excellent fatigue resistance (50 cycles with a set strain of 50%).

## Results and discussion

The assembly and aggregation states of SNWs in 3D space are determined by the interactions between the SNWs and between the SNWs and solvents. Notably, the surface functional groups of SNWs serve as

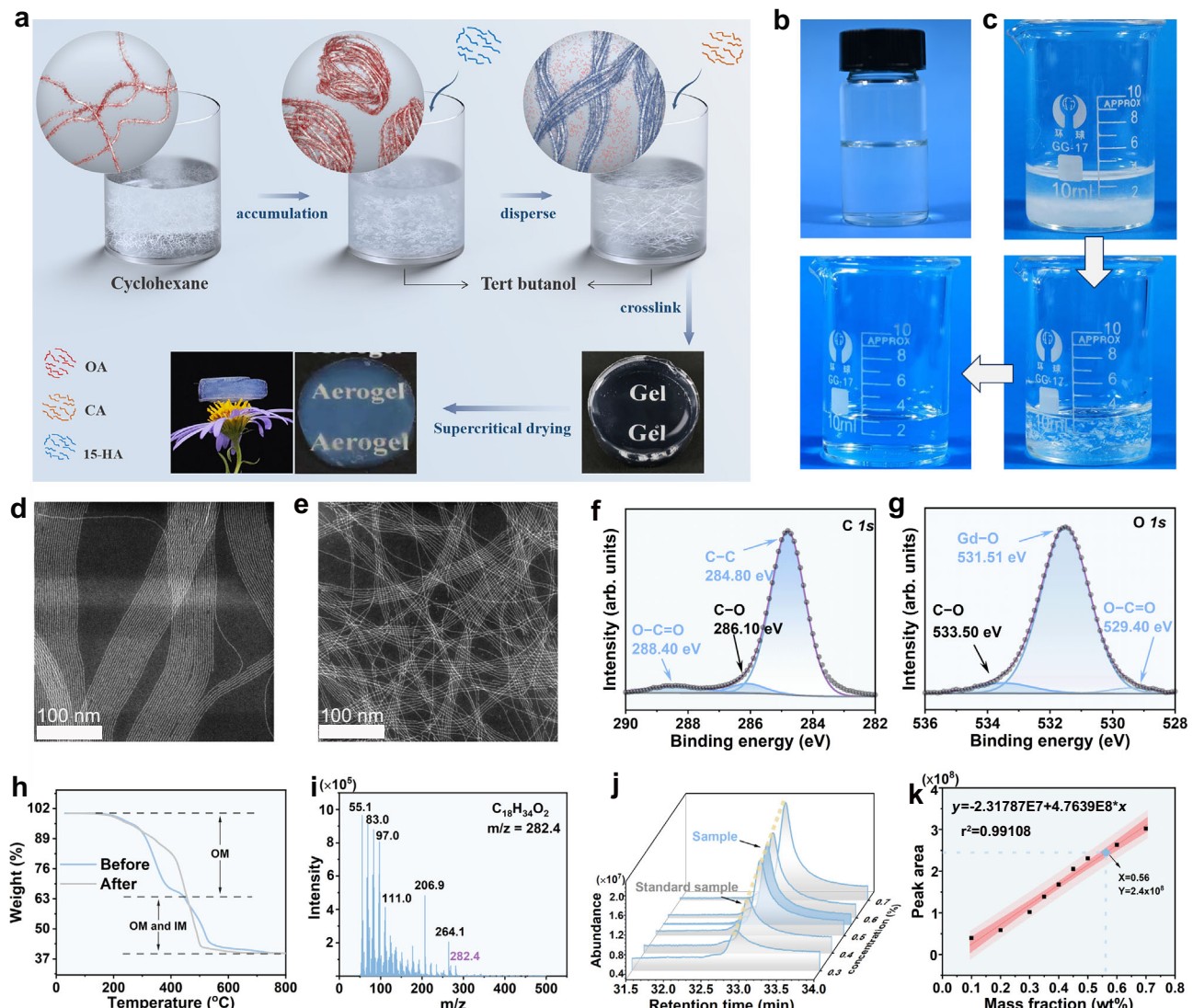

**Fig. 1 | Ligand regulation of Gd–SNWs and characterization of their related properties. a** Flowchart for the ligand regulation of Gd–SNWs and preparation of Gd–SNWAs, where OA, CA, and 15-HA denote oleic acid, citric acid, and 15-hydroxypentanoic acid, respectively. The dispersion of Gd–SNWs in **b** cyclohexane and **c** TBA. STEM images of Gd–SNWs in **d** *n*-octane and **e** TBA, that is, before and after ligand regulation, respectively. High-resolution XPS spectra for the **f** C *1s* and **g** O *1s* peaks in the Gd–SNWs spectrum. **h** Thermal decomposition curves of Gd–SNWs before and after ligand regulation. **i** Mass spectrometry of OA in solvent after ligand regulation. **j** Gas chromatograms and **k** linear regression equation of OA standard samples, with the OA in solvent data point highlighted.

critical determinants that modulate both the interaction modalities. Therefore, by regulating the functional groups on the surface of SNWs, the interactions can be modulated to control the aggregation of SNWs. As shown in Fig. 1b, the Gd−SNWs exhibit excellent dispersibility in cyclohexane, forming a macroscopically transparent dispersion. This is primarily attributed to the abundant oleic acid (OA) ligands anchored on the surface of Gd−SNWs, which provide a compatible interface for the uniform dispersion within non-polar media. However, upon exposure to polar solvents, the solvophobic nature of these surface ligands triggers rapid interwire aggregation, culminating in the precipitation of the Gd−SNWs. We successfully modified the surface of Gd−SNWs with a ligand with a terminal hydroxyl group, which changed the aggregation state of Gd−SNWs in tert-butanol (TBA) solvent (Fig. 1c). In addition, to achieve controlled gels of Gd−SNWs in three-dimensional space, other molecules must be introduced to regulate the interactions between Gd−SNWs. The use of polar solvents facilitated the introduction of crosslinking agents.

Scanning transmission electron microscopy (STEM) revealed that the Gd−SNWs have a highly ordered arrangement in n-octane prior to ligand regulation (Fig. 1d), characterized by a parallel orientation, uniform distribution, and no significant entanglement between the Gd−SNWs. At the same concentration, the morphology of the Gd−SNWs did not change significantly after ligand regulation; however, the Gd−SNWs were interspersed with each other (Fig. 1e). This suggests that the interactions between the Gd−SNWs containing new ligands and between the Gd−SNWs and solvent, may be altered by ligand regulation. Consequently, changes in the repulsive or attractive forces between the Gd−SNWs affect their dispersion state.

X-ray photoelectron spectroscopy (XPS) was used to identify the functional groups of the Gd−SNWs before and after ligand regulation. As shown in Supplementary Fig. S1, the C1s and O1s spectra of the Gd−SNWs before ligand modulation exhibit the characteristic peaks of oleic acid (OA) at 288.5 and 529.8 eV[21]. After ligand regulation, additional C-O characteristic peaks (286.10 and 533.50 eV) appeared[22,23], indicating that the Gd−SNWs contained 15-hydroxypentanoic acid (15-HA) on their surface (Fig. 1f, g and Supplementary Fig. S2). Furthermore, Fourier-transform infrared (FTIR) analysis revealed a distinct change in the peak shape of the hydroxyl stretching vibration band of Gd−SNWs at approximately 3400 cm$^{-1}$ before and after ligand regulation, which is mainly attributed to the effect of 15-HA (Supplementary Fig. S3a, b). The above experimental phenomena indicate that 15-HA was successfully introduced onto the surface of the Gd−SNWs through ligand regulation.

The thermal decomposition curves of the Gd−SNWs were used to determine the proportion of the organic and inorganic components in the Gd−SNWs before and after ligand regulation. As shown in Fig. 1h, the ratio of organic matter (OM) to inorganic matter (IM) in the Gd−SNWs remained unchanged before and after ligand regulation, indicating that the ligand regulation of Gd−SNWs is essentially a ligand displacement process. Further analysis of the thermal decomposition process reveals that most of the OM undergo thermal decomposition in the range of 200–450 °C. As the temperature increases to 450–600 °C, not only will a small amount of OM continue to decompose, but some GdOOH thermally decomposes into Gd$_2$O$_3$[17]. Further analysis showed that ligand regulation did not significantly alter the proportion of OM in the Gd−SNWs, which remained at approximately 43% before and after the regulation (detailed calculations and analysis are provided in the Supplementary Information).

To study the ligand displacement process of organic acids, gas chromatography−mass spectrometry was used to identify organic acids in the solvent after ligand regulation. The mass spectrum of the peak appearing at 32.88 min matches that of OA in the National Institute of Standards and Technical Chemistry (NIST 11.0) database (Fig. 1i and Supplementary Fig. S4). The appearance of OA in the solution indicates that OA on the Gd−SNWs was successfully replaced by 15-HA.

This further suggests that the ligand regulation of Gd−SNWs is a ligand displacement process. To determine the amount of 15-HA substituted for OA, an external standard method (details are provided in the Standard Solution Preparation section in the Supplementary Information) was used to quantify the OA in solution after ligand replacement. As shown in Fig. 1j, k, the mass fraction of OA in the solution was 0.56%. Furthermore, correlation calculations indicated that OA in solution accounted for ~43% of the mass of the Gd−SNWs, which was consistent with the percentage of OM in the Gd−SNWs before ligand regulation (calculations and analysis are detailed in the Supplementary Information). These results showed that OA was almost completely displaced from the Gd−SNWs by 15-HA during ligand exchange.

After ligand regulation, the surface of the Gd−SNWs was enriched in −OH groups, which enabled them to be dispersed in polar solvents, including ethanol, various isomers of butanol, and pentanol (Supplementary Fig. S5b). Notably, Gd−SNWs dispersed in polar solvents aggregated into bundles and exhibited an irregular staggered distribution. In addition, the Gd−SNWs exhibited varying degrees of aggregation in different alcoholic solvents. As shown in Fig. 2a, in the four isomeric solvents of butanol, the aggregation and degree of interlacing of the Gd−SNWs increased as the number of butanol molecular branches increased. This is because the interactions between the different solvents and Gd−SNWs are different. Ultimately, when the Gd−SNWs dispersion system reached equilibrium, its dispersion state changed. Therefore, the controllable aggregation of the Gd−SNWs can be achieved by regulating their dispersion in different alcohol solvents.

When citric acid (CA) was added to the alcohol dispersion of the Gd−SNWs, the Gd−SNWs formed a gel (Supplementary Fig. S6a). Based on STEM image analysis, the microstructural evolution of the four Gd−SNWs butanol gels (SNWGs) corresponded to the evolution of Gd−SNWs dispersed in butanol (Fig. 2a, b). As the number of branched chains of butanol increased, the fiber skeleton of the SNWGs thickened, and the degree of interlacing increased. In addition, the gels of Gd−SNWs assembled in n-butanol (SNWNG), isobutanol (SNWIG), sec-butanol (SNWSG), and tert-butanol (SNWTG) exhibited enhanced strengths (Supplementary Fig. S7). This is because the increased aggregation of Gd−SNWs created a stronger mechanical pathway for stress dissipation, which improved impact resistance.

Molecular dynamics (MD) simulations were used to study the Gd−SNWs gel assembly process in TBA. First, four Gd−SNWs, 30 ionized CA molecules, and 12 H$_3$O$^+$ ions were placed in a closed system with dimensions of $10 \times 10 \times 10$ nm$^3$. The entire system was then solvated using TBA. After running the MD simulations, as time advanced, the Gd−SNWs approached each other and eventually assembled, driven by ionized citrate molecules and protons (Fig. 2c and Supplementary Video S1). As shown in Fig. 2d, the electrostatic interaction energy and van der Waals interaction energy between the Gd−SNWs and ionized CA molecules decreased, which was the main factor driving the assembly between the Gd−SNWs. Smaller fluctuations in the electrostatic forces between the Gd−SNWs were the basis for the assembly (Fig. 2e). The van der Waals forces between the Gd−SNWs improved the stability of the assembled structure. In addition, the interaction energy between the Gd−SNWs and protons fluctuated considerably (Fig. 2f). This strong and unstable electrostatic interaction may change the charge distribution of the Gd−SNWs, which in turn affected the interaction of the Gd−SNWs with the surrounding molecules and contributed to the aggregation of the Gd−SNWs and the assembly of the gel. Thus, the construction of the 3D network of Gd−SNWs is not only the result of strong interactions between the Gd−SNWs and ionized citrate molecules and protons but also of the stable interactions between the Gd−SNWs.

After the Gd−SNWs gels were left standing, SNWNG showed fluidity recovery, while SNWTG remained stable (Fig. 2g). The mechanical strength of SNWG was studied using a localized pressure

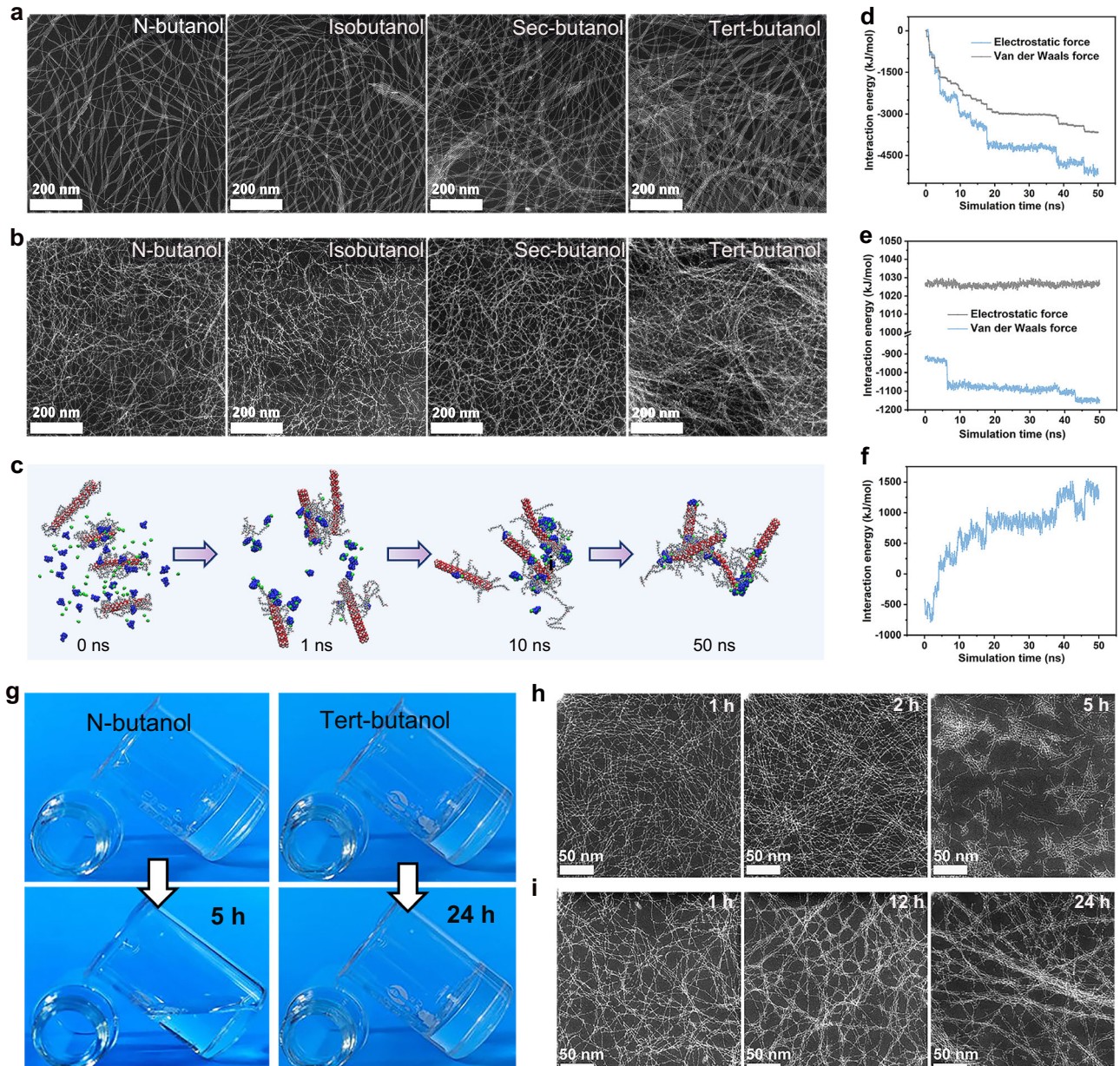

**Fig. 2 | STEM images, MD simulations, and gel structure changes of Gd–SNWs in a butanol system. a** STEM images of Gd–SNWs dispersed in butanol isomers. **b** STEM images of the corresponding Gd–SNWs butanol gels (SNWGs). **c** MD simulations of Gd–SNWs gels in TBA: schematics of Gd–SNWs undergoing assembly. The small blue and green models indicate an ionized CA molecule and a proton, respectively. The interaction energy between **d** the Gd–SNWs and ionized CA molecules, **e** the Gd–SNWs, and (**f**) the Gd–SNWs and protons. **g** SNWGs after standing. STEM images of **h** SNWNG and **i** SNWTG over time.

test. As shown in Supplementary Fig. S8, after the initial formation of SNWG, there was a strength-enhancement phase. With time, when the mechanical strength of SNWG reached its maximum, the strength of SNWNG began to decline, while the strength of SNWTG remained stable. STEM was used to observe the microstructure of the SNWGs at different time points to study the mechanism of mechanical strength changes in the SNWGs. As shown in Fig. 2h, i, the skeletons of SNWNG and SNWTG, respectively, continued to aggregate for ~2 and 12 h. Therefore, the continuous growth of SNWG strength was caused by the further aggregation of Gd–SNWs. As time progressed, the skeleton of the SNWTG stabilized, whereas the skeleton of SNWNG broke, resulting in the strength of SNWNG decreasing and eventual fluidity recovery. This may be because the initial skeleton of SNWG is composed of bundles of Gd–SNWs aggregated in a solvent. The thin bundles formed

by the aggregation of Gd–SNWs in n-butanol resulted in the initial skeleton of SNWNG being too thin. Then, in the subsequent continuous aggregation of Gd–SNWs, the initial skeleton of SNWNG is too thin to withstand the continuous action of the aggregation force, which leads to skeleton collapse and the gel regaining fluidity. In contrast, the initial gel skeleton of SNWTG was strong; therefore, the skeleton remained stable. In addition, when the SNWG was stabilized, the Gd–SNWs exhibited an alternating aggregated–segregated distribution (Fig. 2i), analogous to the transition of cellulose molecular chains between crystalline and amorphous regions. Nevertheless, the extent of Gd–SNWs aggregation rarely surpassed 6 nm. These results indicate that Gd–SNWs undergo further assembly after initial gelation, resembling Ostwald ripening in conventional sol–gel systems. However, beyond a certain time point, the structural damage caused by the

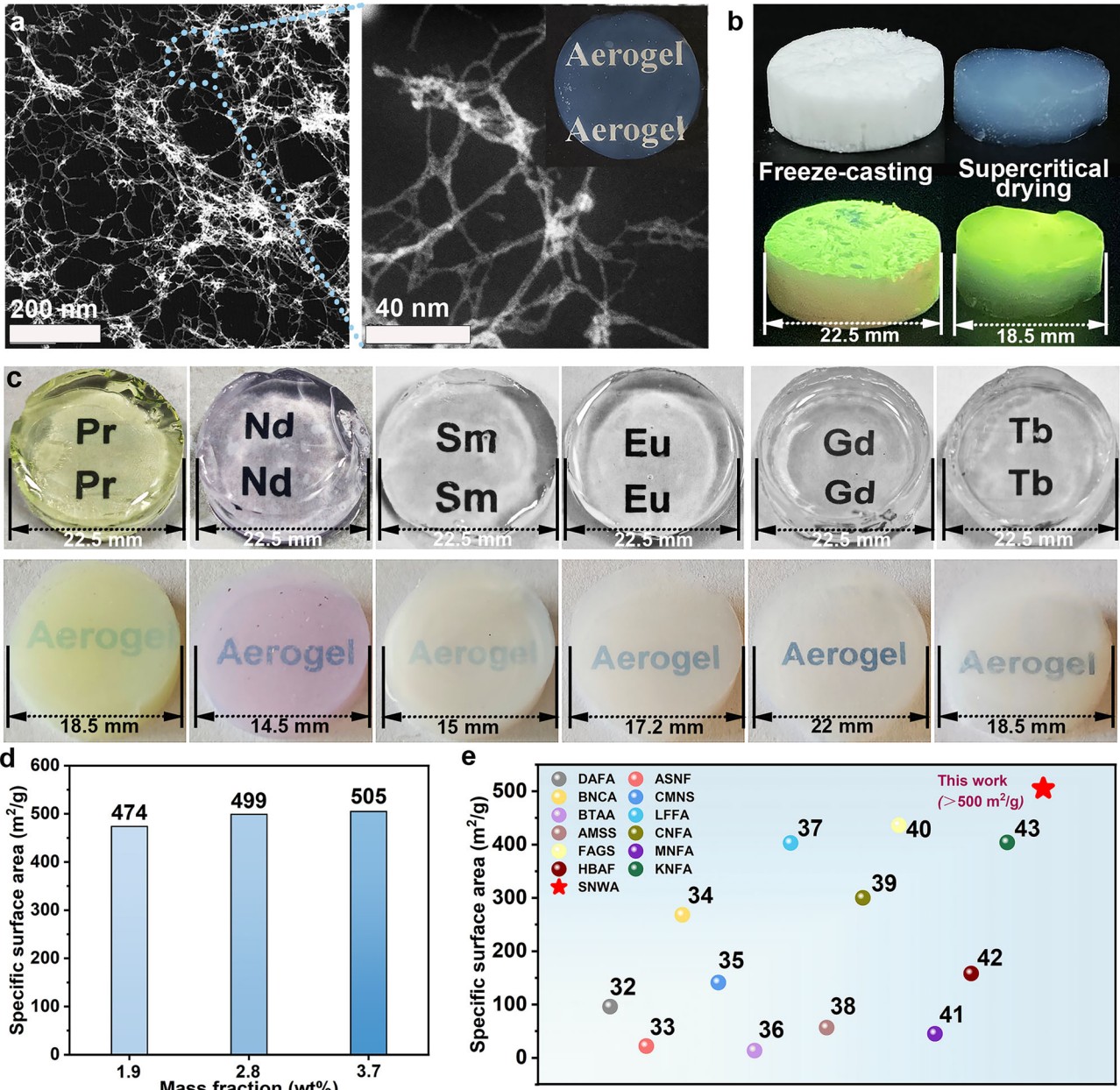

**Fig. 3 | STEM images, MD simulations, and gel structure changes of Gd–SNWs in a butanol system. a** STEM images of 2.8% Gd–SNWA (Gd–SNWA prepared by dispersion with a Gd–SNWs mass fraction of 2.8%). The inset is an optical photograph of Gd–SNWA on paper. **b** Gd–SNWAs prepared by freeze-casting (left) and supercritical drying (right), and their state under UV light. **c** Photographs of six kinds of rare-earth oxide gels and their aerogels. **d** Specific surface area of Gd–SNWAs. **e** Comparison of the specific surface area of Gd–SNWAs and other nanofiber aerogels[32–43].

fracturing of Gd–SNWs outweighs the mechanical enhancement from aggregation, leading to an overall degradation in the gel properties. When the Gd–SNWs broke more severely, the gel regained fluidity.

Gd–SNWAs were obtained by removing the internal liquid Gd–SNWG via supercritical drying. The Gd–SNWAs appeared to be in a semi-translucent state. When the Gd–SNWA is placed over the word "Aerogel" printed on paper, the text is clearly visible (Fig. 1a and the inset in Fig. 3a), indicating that it has a certain transmittance to visible light. The microstructure of Gd–SNWA consisted of an ultrafine network formed by Gd–SNWs fibers with diameters ranging from ~2 to 6 nm (Fig. 3a). Some ligands on the Gd–SNWs surface were removed during supercritical drying, which reduced their steric hindrance effect (Supplementary Fig. S9) and enhanced the interaction force between the Gd–SNWs. Therefore, the Gd–SNWs tend to aggregate to form an

integrated structure, rather than exhibit the morphological features of individual Gd–SNWs. In addition, because aggregated Gd–SNWs are stronger than individual Gd–SNWs, the network skeleton formed by aggregated Gd–SNWs can withstand environmental changes without collapsing during the drying process.

As the concentration of the Gd–SNWs increased, the transparency of Gd–SNWA gradually decreased (Supplementary Fig. S10). This was attributed to a decrease in the homogeneity of the dispersion of the Gd–SNWs with increasing concentration, which reduced the light transmittance of the aerogel. The UV-vis spectra of the Gd–SNWAs enabled a clearer observation of this phenomenon (Supplementary Fig. S11). In contrast to the white TbOOH SNWAs (Tb–SNWA) obtained in our previous freeze-casting study, the transparent Tb–SNWA obtained here by the controlled assembly of TbOOH SNWs (Tb–SNWs)

followed by supercritical drying has significantly better photo-luminescence (PL) properties. As shown in Fig. 3b, when irradiated with UV light, both Tb–SNWAs emitted yellow-green light owing to the excitation of Tb$^{3+}$ by the UV light (at an excitation wavelength of 254 nm). Notably, the translucent Tb–SNWA prepared via the controlled assembly of Tb–SNWs by supercritical drying exhibited a significantly enhanced luminescence brightness compared to the white Tb–SNWA obtained via directional freeze-drying. Specifically, the luminescence can be observed throughout the entire bulk material rather than being confined merely to the surface. This phenomenon originates from the ultrafine nanofiber network and highly homogeneous nanoporous architecture achieved through controllable assembly and supercritical drying. Notably, the fiber diameter of this ultrafine fibrous network (~6 nm) is far smaller than the wavelength of visible light. This unique characteristic not only facilitates the deep penetration of UV light into the material but also enables the lateral transmission of the emitted PL, thereby achieving volume excitation. Conversely, the inhomogeneous structure inherent to freeze-cast materials induces intense surface scattering, which significantly restricts the penetration depth of the excitation light, ultimately resulting in faint surface-confined luminescence.

To elucidate the optical performance of Tb–SNWA, steady-state PL emission and excitation spectra were recorded at room temperature (Supplementary Fig. S12a). The PL emission spectra exhibit the characteristic emission fingerprint of Tb$^{3+}$. Four distinct emission bands were observed at 490, 545, 585, and 622 nm, which were assigned to the radiative transitions from the $^5D_4$ excited state to multiple $^7F_J$ (J = 6, 5, 4, and 3, respectively) ground-state manifolds. Notably, the hypersensitive transition at 545 nm ($^5D_4$–$^7F_5$) dominates the spectrum, accounting for the high-purity green luminescence observed macroscopically. The narrow full-width at half maximum of these peaks underscores the excellent monochromaticity of Tb–SNWA. The PL excitation spectra, obtained by monitoring the green emission at 545 nm, exhibit a series of sharp peaks in the near-UV region (300–400 nm). These features correspond to the intra-configurational 4f–4 f transitions of Tb$^{3+}$, such as $^7F_6$–$^5D_{2,3}$ and $^7F_6$–$^5L_{10}$. Furthermore, the broad excitation band observed in the deep-UV region (<300 nm) was attributed to the 4f–5d transitions of Tb$^{3+}$. This wide excitation range facilitates efficient energy harvesting under standard UV illumination, confirming the potential of Tb–SNWA for advanced optoelectronic applications.

In addition, other rare-earth SNWs can also be used as building blocks to assemble M–SNWAs (Fig. 3c), where M refers to rare-earth metals such as Pr, Nd, Sm, Eu, Gd, and Tb, indicating the generality of this assembly strategy. Furthermore, different types of rare-earth SNWs can be blended and assembled together. For example, EuOOH SNWs (Eu–SNWs) and Tb–SNWs were blended and assembled, followed by supercritical drying to obtain Eu/Tb–SNWAs with tunable PL emission (Supplementary Fig. S12). By increasing the ratio of Tb–SNWs to Eu–SNWs in the blend, the light emitted by the aerogel transitions from bright red and light orange to light green owing to the enhanced green emission.

The nitrogen adsorption–desorption isotherms of the Gd–SNWAs were type IV isotherms with H1 type hysteresis loops (Supplementary Fig. S13a). This indicates that a mesoporous structure was formed inside the Gd–SNWAs. As the concentration of the Gd–SNWs increased, the amount of nitrogen adsorbed by the Gd–SNWAs also increased. This is because, as the concentration of Gd–SNWs increases, the percentage of Gd–SNWs with lower aggregation increases, and the aggregation becomes more disorganized, which leads to the original larger pores being subdivided into mesopores. The Barrett–Joyner–Halenda (BJH) pore size distribution of the Gd–SNWAs further confirmed this (Supplementary Fig. S13b). The mesoporous structure of the Gd–SNWAs became more prominent with increasing Gd–SNWs concentration. Consequently, the specific surface area of

the Gd–SNWAs increased with increasing Gd–SNWs concentration (Fig. 3d). The Gd–SNWA synthesized via supercritical drying achieved a remarkably high specific surface area of 505 m$^2$ g$^{-1}$, marking a substantial leap compared to previously reported freeze-cast SNWAs (20 m$^2$ g$^{-1}$)[17]. This pronounced enhancement is ascribed to the effective preservation of the intrinsic ultrafine architecture of the Gd–SNWs during the supercritical drying process. In contrast, the ice crystal growth during freeze-casting inevitably exerts compressive forces that squeeze the SNWs and collapse the pore structure, ultimately leading to a dramatic reduction in surface area. Furthermore, aerogels constructed from traditional nanoscale nanowires have a specific surface area of 10–436 m$^2$ g$^{-1}$ (Fig. 3e). In contrast, aerogels assembled using SNWs have a much larger specific surface area due to the ultrafine characteristics of the framework formed by the aggregation of SNWs. When these aggregates interweave, they form a more complex and richer pore network, which leads to more exposed surfaces inside the aerogel, resulting in a larger specific surface area.

In addition, the ultrafine network skeleton of Gd–SNWA results in a significantly lower density (0.024 g cm$^{-3}$) than conventional metal oxide aerogels, such as CuO (0.267 g cm$^{-3}$)[24], Al$_2$O$_3$ (0.078–0.106 g cm$^{-3}$)[25], Fe$_2$O$_3$ (0.26–0.30 g cm$^{-3}$)[26], ZrO$_2$ (0.16 g cm$^{-3}$)[27], and Y$_2$O$_3$ aerogels (0.15 g cm$^{-3}$)[28]. The Gd–SNWA placed on a flower displayed lightweight characteristics (Fig. 1a).

Because the slender skeleton assembled by Gd–SNWs is weak, the Gd–SNWAs are prone to deformation when impacted by external forces (Supplementary Video S2), which limits the use of aerogels in certain situations and causes adverse effects. Therefore, we adopted chemical vapor deposition (CVD) to enhance the mechanical properties of the Gd–SNWAs using methyltrimethoxysilane (MTMS) and H$_2$O. As shown in Fig. 4a, the skeleton of the aerogel was covered by rigid silicon dioxide-containing methyl groups after CVD modification, and the aerogel appeared white. XPS survey spectra were used to verify whether the Gd–SNWAs were successfully modified by silanization. As shown in Fig. 4b, the spectrum of Gd–SNWA after silanization modification has an extra Si 2p characteristic peak at 102 eV, which indicates the presence of silicon in the aerogel. The high-resolution XPS spectra of the Si 2p peak was deconvoluted into two peaks at 102.9 and 104.7 eV corresponding to Si–O and Si–C (Supplementary Fig. S14)[29,30], respectively, which indicated that silicon grafting on the Gd–SNWA and silanization modification were successful.

SEM analysis revealed that Gd–SNWA has a porous network structure before and after silanization and there was no significant change in the diameter of the fiber backbone (Fig. 4c), meaning that Gd–SNWA still has a low density (0.041 g cm$^{-3}$) after being modified (Fig. 4d). The energy-dispersive X-ray spectroscopy (EDS) spectra of Gd–SNWA showed additional silicon peaks after silanization (Fig. 4e and Supplementary Fig. S15a), which further proved the presence of silicon in the aerogel after silanization. In addition, the EDS elemental distribution map shows that silicon is uniformly distributed on the surface of the skeleton of Gd–SNWA, which proves that the fiber skeleton of Gd–SNWA is covered by a layer of silica-containing methyl groups.

The compressive stress–strain curves of Gd–SNWA prepared with different concentrations of Gd–SNWs are shown in Supplementary Fig. S16. The compressive strength of Gd–SNWA improved nearly tenfold after silanization. In addition, the stress–strain curves of Gd–SNWA after loading–unloading cycles showed that the aerogels before modification had poor resilience and broke after compression (Fig. 4g and Supplementary Video S2). After silanization, Gd–SNWA still exhibited nearly 100% elastic recovery after 50 loading–unloading cycles of compression (Fig. 4h and Supplementary Video S3). This excellent elastic property was attributed to the rigid layer consisting of Si–CH$_3$ and Si–O–Si (Supplementary Fig. S17) covering the skeleton of the Gd–SNWA, which endowed the Gd–SNWA with excellent

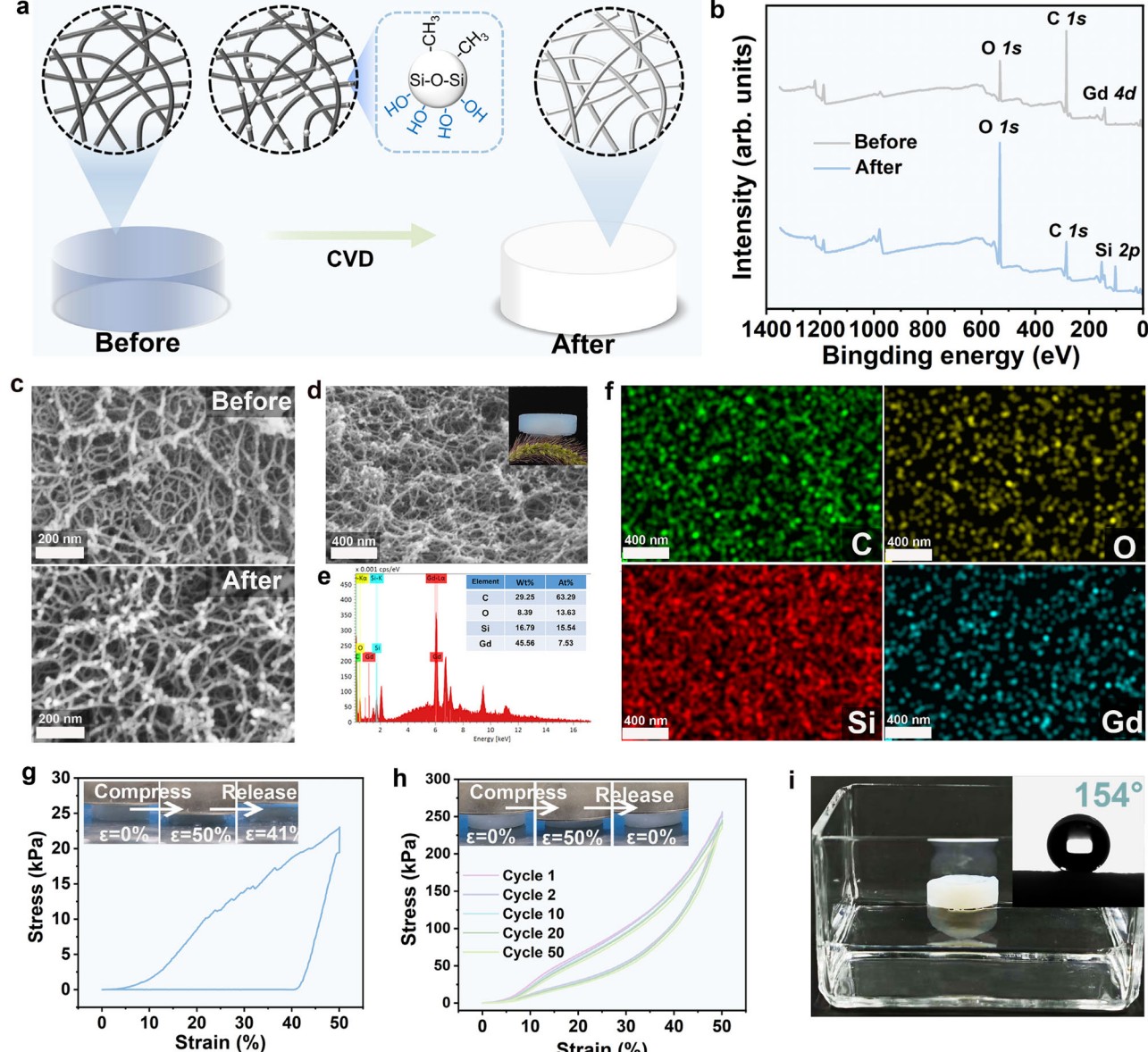

**Fig. 4 | Silanization process of Gd–SNWA and characterization of their properties and structure before and after silanization. a** Silanization modification process of Gd–SNWA. **b** XPS survey spectrum and **c** SEM image of Gd–SNWA before and after silanization. **d** SEM image, **e** elemental mapping, and **f** elemental distribution of Gd–SNWA modified by silanization. **g** Compression stress–strain curve of Gd–SNWA. The insets show optical photographs of an aerogel compressed once. **h** Stress–strain curves of 50 compression curves of Gd–SNWA modified by silanization. The inset shows an optical photograph of an aerogel compressed 50 times. **i** Optical photograph of Gd–SNWA modified by silanization floating on water. The inset shows the water contact angle of Gd–SNWA.

compressive resistance. In addition, the existence of these rigid layers endows the Gd–SNWA with excellent hydrophobic properties (water contact angle of 154°), which enables Gd–SNWA to resist water vapor erosion in humid environments (Fig. 4i).

In this study, the interactions between Gd–SNWs and between Gd–SNWs and solvents were modulated by adjusting the surface ligands of the Gd–SNWs. This approach successfully controlled the aggregation state of the Gd–SNWs in polar solvents, achieving effective dispersion. On this basis, the controllable assembly of Gd–SNWs in alcohol solvents was realized, and metal oxide aerogels with high specific surface areas and low densities were obtained, which effectively enriched the building blocks and construction methods of the aerogels. After silanization post-treatment, the hydrophobicity and mechanical properties of the aerogels were further enhanced, indicating great potential for various applications. The Gd–SNWAs

exhibits ultralow density (~0.024 g cm$^{-3}$), high specific surface area (505 m$^2$ g$^{-1}$), superior light transmittance, excellent strain-recoverable compressibility, and excellent fatigue resistance (50 cycles with a set strain of 50%). In summary, the strategy of ligand replacement and controllable assembly not only enriches the existence states of SNWs but also offers a novel strategy for constructing high-performance aerogels.

## Methods
### Experimental materials
Oleic acid (C$_{18}$H$_{34}$O$_2$), Oleylamine (C$_{18}$H$_{37}$N), Gadolinium chloride hexahydrate (GdCl$_3$·6H$_2$O), Methyltrimethoxysilane (CH$_3$Si(CH$_3$O)$_3$), Citric acid (C$_6$H$_8$O$_7$) and 15-hydroxyeicosenoic acid (C$_{15}$H$_{30}$O$_3$) are obtained from Aladdin. N-butanol (C$_4$H$_{10}$O), Isobutanol (C$_4$H$_{10}$O), Secondary butanol (C$_4$H$_{10}$O), and tert-butanol (C$_4$H$_{10}$O) are obtained

from Macklin. Ethanol absolute ($C_2H_5OH$) and cyclohexane ($C_6H_{12}$) are obtained from Tianjin Bohua Chemical Reagents.

## Synthesis of REOOH sub-1 nm nanowires

REOOH (RE=Pr, Nd, Sm, Eu, Gd, Tb) sub-1 nm nanowires (SNWs) were synthesized through a hydrothermal synthesis method. Here, we took the synthesis of GdOOH sub-1 nm nanowires (Gd−SNWs) as an example. 0.8 g $GdCl_3 \cdot 6H_2O$ was dispersed in 12 mL $C_2H_5OH$ and 1 mL deionized water. Then added into the mixed dispersion of 4 mL oleylamine (OM) and 2 mL oleic acid (OA) with stirring. Next, the reactants were placed in a 50 mL reactor using 170 °C heating for 4 h. After the reaction, the viscous dispersion was dispersed with cyclohexane and precipitated by centrifugation with $C_2H_5OH$. The supernatant was then poured off, and the precipitate was again dispersed using cyclohexane and centrifuged using $C_2H_5OH$ to purify the SNWs. For the synthesis of other rare-earth SNWs, the amounts of precursors were determined based on the molar ratio used for $GdCl_3 \cdot 6H_2O$, while maintaining identical solvent volumes across all preparations. All SNWs in this article were prepared according to the method previously reported by Xun Wang and co-workers[11,31].

## Ligand regulation of sub-1 nm nanowire

Here, we took GdOOH sub-1 nm nanowires (Gd−SNWs) as an example. Use tert-butanol (TBA) to precipitate SNWs from the dispersion and centrifuge at 10,000 r/min. Then, clean the precipitated SNWs three times with TBA and air dry. Stir 0.075 g SNWs in 3.2 mL TBA and add 0.04 g 15-hydroxyeicosenoic acid. After 30 min, SNWs were completely dispersed in tert-butanol solvent and the dispersion appeared transparent.

## Preparation of sub-1 nm nanowire aerogel

About 0.1 mL of citric acid ethanol solution (mass fraction 38.8%) was added to the TBA dispersion of the above SNWs. After stirring for 10 s, the TBA dispersion of SNWs will form a transparent gel. Then, transparent gel was supercritical dried to prepare semi-transparent sub-1 nm nanowire aerogels (SNWAs).

## Supercritical drying process

Aerogel fabrication was conducted via a supercritical $CO_2$ drying apparatus, and the detailed operational procedures were as follows. First, the extraction reactor was preheated to 45 °C. After loading the sample, the internal pressure of the reactor was increased to 17 MPa at a rate of 0.1 MPa per 30 s, followed by a constant-pressure holding step at 17 MPa for 5 h to maintain pressure stability. Upon completion of the pressure holding, two sequential heating steps were performed: the temperature was first raised to 55 °C and held constant for 30 min, and then further increased to 65 °C with another isothermal holding for 30 min. After the heating process, the $CO_2$ circulation was initiated and maintained until no liquid effluent was detected. Finally, the extraction reactor was depressurized to 0 MPa at a controlled rate of 0.1 MPa per 30 s, thus completing the entire drying process.

## Chemical vapor deposition

Two 5 mL centrifuge tubes, each containing 1 mL of methyltrimethoxysilane (MTMS) and 0.8 mL of deionized water, were placed alongside the aerogel samples inside a 1 L desiccator. The desiccator was then hermetically sealed and maintained at 70 °C for 6 h.

## Standard solution preparation

Preparation of a mixed standard solution I: A mixed standard solution of oleic acid (7 wt%) was prepared using a binary solvent system of acetonitrile and tert-butanol with a volume ratio of 15:3.2. The resulting solution was subsequently stored in a sealed glass vial for further use.

Preparation of a working solution: The initial stock solution I was step-wise diluted with the acetonitrile/tert-butanol co-solvent to yield a series of oleic acid standard solutions with a mass fraction gradient of 0.1, 0.2, 0.3, 0.35, 0.4, 0.45, 0.5, 0.6, and 0.7 wt%. These solutions were stored in sealed vials and subsequently utilized for the construction of a calibration curve.

## Characterization

The chemical analysis of the aerogels before and after ligand replacement and after silane modification was carried out using XPS. The thermal weight loss of aerogels in air was studied by using the STA449 F3 thermogravimetric analyzer of NETZSCH. The samples were heated from room temperature to 800 °C at a heating rate of 10 °C/min under a mixed atmosphere of $N_2$ and $O_2$. Samples were prepared by pressing the samples into thin slices. Attenuated total reflection Fourier-transform infrared (ATR−FTIR) spectroscopy of the samples were obtained using PerkinElmer Spectrum 3 in the wavelength range of 500 − 4000 $cm^{-1}$. SNWs and SNWAs were treated separately on ultrathin carbon films and observed by STEM with a Thermo Fisher Talos F200i. Microstructural analysis was performed using a Thermo Fisher Apreo 2C scanning electron microscope. Samples were prepared by placing them on a conductive adhesive and sprayed with conductive coating. The test voltage was 10 KV and the magnification was 10,000 and 50,000 times. Place the cylindrical gel and aerogel on the sample table of the electronic universal testing machine (HD−B609B−S, Dongguan Haida Instrument Co., Ltd). Then, contact the compression tool with the surface of the sample. Perform a compression test on the sample using a compression program, with a testing rate set at 6 mm/min. The aerogels were tested for contact angle using the SZ−CAMC31 Contact Angle Meter (Shanghai Xuanzhun Instrument Co., Ltd.). Place the aerogel on the sample stage, use a microsyringe to drop a deionized aqueous suspension (suspension volume is 6 μL), and slowly contact the surface of the sample to be measured by raising the sample stage. Afterwards, the images were captured using an image sensor and analyzed using contact angle analysis software. In this paper, the contact angle was measured as an average at different points of the aerogel, with at least three measurement points, within ±0.5°. In this paper, the Cary 5000 UV-Vis-NIR spectrophotometer was used for transmittance testing. Select a scanning range of 200 to 800 nm for testing. This experiment used Micromeritics (ASAP 2460 model) to calculate the specific surface area of the sample. Use an Agilent 7000C gas chromatography−mass spectrometer to qualitatively and quantitatively analyze the OA content in the solution after ligand displacement.

## Data availability

All data supporting the findings of this study are available within the paper and its Supplementary files. Any additional information related to the study is available from the corresponding author upon request. The source data generated in this study are provided in the Source Data file. The raw data are available on Zenodo at https://doi.org/10.5281/zenodo.18811572. Source data are provided with this paper.

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

## Acknowledgements

This research was funded by the National Natural Science Foundation of China (52164013 and 22466028) to H.S. and R.F., Local Science and Technology Development Fund Projects Guided by the Central Gov-ernment (2023ZY0022) to H.S., the Natural Science Foundation of Inner Mongolia (2025YQ047) to H.S.

## Author contributions

Y.D., Y.X., and H.S. conceived and designed the research. Y.D. carried out the experiments, analyzed data and wrote the manuscript. X.Y., R.F., D.L., and L.L. assisted in the characterization. Y.D., Y.X., R.F., and H.S. supervised the project and modified the manuscript. All authors dis-cussed the results and reviewed the manuscript.

## Competing interests

The authors declare no competing interests.
