## [Transparent Peer Review file · Nature Communications]

Controllable assembly of sub-1 nm nanowires for the construction of aerogels

Corresponding Author: Professor Huazheng Sai

Version 0:

Reviewer comments:

Reviewer #1

(Remarks to the Author)

This manuscript achieves the controllable assembly of sub-1 nm nanowires in three-dimensional space through ligand regulation and solvent polarity control. Metal oxide aerogels with low density, high specific surface area, good light transmittance, and excellent compressive resilience have been successfully constructed. This research enriches the construction strategy of aerogel materials and shows significant innovation and application potential. I recommend that this manuscript be accepted after addressing the following issues.

1. The supplementary Figure S4 mentioned in the manuscript is inconsistent with the supplementary Figure S4 in the supporting information. The authors are requested to verify this.
2. The manuscript mentions that SNWs can be dispersed in polar solvents such as ethanol and butanol isomers after ligand regulation, but does not show a direct dispersion comparison of SNWs in these solvents before and after ligand substitution.
3. The author should compare the advantages of SNWs aerogels obtained by supercritical drying compared with SNWs aerogels obtained by freeze casting.
4. Has the specific surface area of the aerogel changed after gas-phase modification?
5. The manuscript mentioned chemical vapor deposition modified SNWs aerogels, but did not describe the relevant experimental details.

Reviewer #2

(Remarks to the Author)

This manuscript reports an intriguing study that employs ultrafine nanowires to construct aerogels, thereby achieving further miniaturization of the aerogel building blocks. The noteworthy results include the successful fabrication of sub-1 nm nanowire-based aerogels with ultralow density, high specific surface area, excellent optical transparency, and remarkable compressive resilience. This work is significant to the fields of porous materials and lightweight functional materials, providing a promising strategy for the rational design of high-performance aerogels and advancing beyond conventional nanoparticle or cluster-based aerogels. Overall, this work is of high quality, and I suggest acceptance after minor revisions addressing the following comments.

1. The preparation method of the SNWs was not introduced either in the main text or in the SI. In particular, the authors should clearly state what specific SNWs are investigated in this work (e.g., Gd_2O_3 ?). This information should be explicitly provided in the manuscript and in the corresponding figure captions.
2. Figures 1b and 1c are presented without sufficient discussion or explanation of the observed phenomena. The authors should elaborate on the key findings illustrated by these figures to enhance the reader's understanding.
3. The authors employ several characterization techniques to demonstrate the regulation of surface ligands on SNWs. However, solvent-related peaks can still appear even if the SNWs were merely immersed in those solvents. It is therefore recommended that the authors provide additional spectral evidence, such as characteristic peak shifts, to substantiate the proposed surface ligand regulation.
4. The manuscript discusses the photoluminescent properties of the aerogels, yet the excitation wavelength used in the measurements has not been specified. As photoluminescence is highly dependent on the excitation wavelength, this

information is essential. Moreover, did the authors do the quantized photoluminescence measurement?

5. The authors report modifying the SNW aerogels via CVD using MTMS and water; however, the specific processing parameters are not described. Similarly, key conditions for the supercritical drying process such as the type of supercritical fluid, drying temperature, and pressure are missing. These parameters critically influence the pore structure and density of the final aerogels, and are thus necessary for result interpretation and experimental reproducibility. The complete preparation and drying parameters should be included.

6. On page 7, the phrase “an external standard method” is used but not defined. The authors should clarify what this method refers to and provide appropriate references if applicable.

7. Certain details have been overlooked, which impede comprehension. For example, the label “15-HEA” in Figure 1a is undefined and not explained in the text or figure caption. The authors should ensure that all figure labels are clearly defined to facilitate accurate interpretation by readers. Besides, a spelling error is present in the label of the horizontal axis in Figure S11, which should be corrected to “Wavelength”.

8. The final section titled “Discussion” should be renamed “Conclusion.” In addition, the writing throughout the manuscript need to be further polished to improve clarity, consistency, and overall readability.

Version 1:

Reviewer comments:

Reviewer #1

(Remarks to the Author)

I have reviewed the revised manuscript and the author's response letter. The author responded carefully to the questions I raised. The manuscript is now clearer and the conclusion is more fully supported. Therefore, I suggest accepting it.

Reviewer #2

(Remarks to the Author)

All my previous comments have been addressed. I would only suggest a minor correction: the horizontal axis in Figures S12a and S12b should be labeled as “Wavelength.” In addition, the values of 545 nm and 615 nm indicated in the PLE curves in Figures S12a and S12b appear to correspond to the emission wavelengths rather than the excitation wavelength.

Response Letter to Reviewers

We appreciated the editor for permitting us to revise our manuscript. After careful modification along the reviewer's suggestions, we submit the revised manuscript and the response to these comments now.

Response to the reviewer's comments:

Thanks for the helpful comments which are of great help to improve the quality of the manuscript. The changes in the revised manuscript have been marked with a yellow background.

Comments:

Reviewer: 1

Remarks to the Author

This manuscript achieves the controllable assembly of sub-1 nm nanowires in three-dimensional space through ligand regulation and solvent polarity control. Metal oxide aerogels with low density, high specific surface area, good light transmittance, and excellent compressive resilience have been successfully constructed. This research enriches the construction strategy of aerogel materials and shows significant innovation and application potential. I recommend that this manuscript be accepted after addressing the following issues.

(1) The supplementary Figure S4 mentioned in the manuscript is inconsistent with the supplementary Figure S4 in the supporting information. The authors are requested to verify this.

Answer:

We sincerely apologize for this citation error and thank the reviewer for spotting the mismatch. You are absolutely correct; the data described in this section corresponds to **Supplementary Fig. S5, not Fig. S4** as erroneously stated in the original manuscript.

We have corrected the citation in the revised manuscript to correctly refer to Supplementary Fig. S5. We have also carefully cross-checked all figure citations in the main text against the Supplementary Materials to ensure accuracy.

The revised text has been incorporated into the first paragraph of page 9, as follows:

After ligand regulation, the surface of the Gd-SNWs was enriched in -OH groups, which enabled them to be dispersed in polar solvents, including ethanol, various isomers of butanol, and pentanol (Supplementary Fig. S5b).

Supplementary Fig. S5 (b) Dispersions of SNWs in ethanol, n-butanol, isobutanol, sec-butanol,

tert-butanol, and 1-pentanol after ligand exchange.

Supplementary Fig. S4 presents the gas chromatography (GC) analysis of the ligands recovered from the solvent following ligand exchange. In conjunction with the mass spectrometry (MS) data shown in Fig. 1i, these results confirm the identity of the ligands as oleic acid.

The relevant analysis has been incorporated into the second paragraph of page 7, as follows:

To study the ligand displacement process of organic acids, gas chromatography–mass spectrometry was used to identify organic acids in the solvent after ligand regulation. The mass spectrum of the peak appearing at 32.88 min match that of OA in the National Institute of Standards and Technical Chemistry (NIST 11.0) database (**Supplementary Fig. S4 and Fig. 1i**).

Supplementary Fig. S4 Gas chromatography of OA in solvent after ligand substitution.

Fig. 1 (i) Mass spectrometry of OA in solvent after ligand regulation.

(2) The manuscript mentions that SNWs can be dispersed in polar solvents such as ethanol and butanol isomers after ligand regulation, but does not show a direct dispersion comparison of SNWs in these solvents before and after ligand substitution.

Answer:

We greatly appreciate the reviewer's constructive suggestion. We agree that a direct comparison is essential to visually demonstrate the improved dispersibility.

It is worth noting that before ligand regulation, the pristine SNWs (capped with oleic acid)

exhibit similar insolubility across these polar solvents (including ethanol and various butanol isomers) due to the hydrophobic nature of the long alkyl chains. Originally, we did not include images of the pristine SNWs in each solvent because their states were identical (i.e., immediate precipitation).

Following your suggestion, we have now included digital photographs of the unmodified SNWs in these solvents **in the revised Supplementary Fig. S5**.

Supplementary Fig. S5 Dispersions of SNWs in ethanol, n-butanol, isobutanol, sec-butanol, tert-butanol, and 1-pentanol (a) before and (b) after ligand exchange.

(3) The author should compare the advantages of SNWs aerogels obtained by supercritical drying compared with SNWs aerogels obtained by freeze casting.

Answer:

We sincerely thank the reviewer for this valuable suggestion. In the revised manuscript, we have added a detailed comparison between the supercritical drying (SCD) Gd-SNWAs and the freeze casting (FC) counterparts. Specifically, we highlighted that the SCD-Gd-SNWAs exhibit a significantly higher specific surface area ($505 \text{ m}^2 \text{ g}^{-1}$ vs. $20 \text{ m}^2 \text{ g}^{-1}$)¹ and superior optical translucency due to the preservation of the ultra-fine nanowire network and uniform nanopores.

The supplementary content has been incorporated into the first paragraphs of page 14 and page 16, respectively, as follows:

Notably, the translucent Tb-SNWA prepared via controlled assembly of SNWs with supercritical drying exhibited a significantly enhanced luminescence brightness compared to the white Tb-SNWA obtained by directional freeze-drying. Specifically, the luminescence can be observed throughout the entire bulk material rather than being confined merely to the surface. This phenomenon originates from the ultrafine nanofiber network and highly homogeneous nanoporous architecture achieved through controllable assembly and supercritical drying. Notably, the fiber diameter of this ultrafine fibrous network (approximately 6 nm) is far smaller than the wavelength of visible light. This unique characteristic not only allows ultraviolet light to penetrate deep into the material but also enables the emitted photoluminescence to transmit laterally, thereby achieving volume excitation. Conversely, the inhomogeneous structure inherent to freeze-cast materials induce intense surface scattering, which significantly restricts the penetration depth of the excitation light, ultimately resulting in merely faint surface-confined luminescence.

Fig. 3 (b) SNWAs prepared by freeze-casting (left) and supercritical drying (right) and their state under UV light.

In this work, the Gd-SNWA synthesized via supercritical drying achieved a remarkably high specific surface area of $505 \text{ m}^2 \text{ g}^{-1}$ (**Fig. 3d**), marking a substantial leap compared to previously reported freeze-cast SNWAs (merely $20 \text{ m}^2 \text{ g}^{-1}$)¹. This pronounced enhancement is ascribed to the effective preservation of the intrinsic ultrafine architecture of the SNWs during the supercritical drying process. In contrast, the ice crystal growth during freeze-casting inevitably exerts compressive forces that squeeze the SNWs and collapse the pore structure, ultimately leading to a dramatic reduction in surface area.

Fig. 3 (d) The specific surface area of Gd-SNWAs obtained through supercritical drying.

(4) Has the specific surface area of the aerogel changed after gas-phase modification?

Answer:

We thank the reviewer for this insightful question. In the revised manuscript, we have included the surface area data before and after modification. After the gas-phase silane modification, the specific surface area of the Gd-SNWA decreased from **505 m² g⁻¹ to 379 m² g⁻¹**. This is because partial silica deposits on the fiber surface during the modification process, and a small number of pores are filled accordingly, which ultimately leads to a decrease in the specific surface area of the material. The relevant nitrogen adsorption-desorption isotherms and the comparison of specific surface areas have been added to the **revised Supporting Information as Supplementary Fig. S13**.

Supplementary Fig. S13 Analysis of specific surface area and pore size of aerogels. (a) Nitrogen

adsorption-desorption isotherms and (b) Barrett-Joyner-Halenda (BJH) pore size distribution of Gd-SNWA. (c) Nitrogen adsorption-desorption isotherms, (d) BJH pore size distribution, and (e) **specific surface area of silane-modified Gd-SNWA.**

A detailed analysis is provided as follows:

After silane modification, the nitrogen adsorption-desorption isotherm of Gd-SNWAs was identified as Type IV with a Type H1 hysteresis loop (Supplementary Fig. S13c), which confirms the preservation of a mesoporous structure within Gd-SNWAs. A comparison of the pore size distributions of SNWAs before and after silane modification revealed a remarkable structural variation. First, a substantial reduction in the incremental pore volume was observed for all modified samples (Supplementary Fig. S13d), which directly demonstrates that silica species were deposited on the fibrous framework and partially blocked the internal void spaces. Second, the pore size range of modified SNWAs exhibited a slight shift toward larger values: the pore size peak of pristine SNWAs was centered at approximately 15–20 nm, whereas that of modified SNWAs shifted to the range of 25–40 nm. This phenomenon is attributed to the preferential blockage of smaller mesopores. These results are in excellent agreement with the observed reduction in the specific surface area of the aerogel after silane modification, which decreased from 505 m² g⁻¹ to 379 m² g⁻¹.

(5) The manuscript mentioned chemical vapor deposition modified SNWs aerogels, but did not describe the relevant experimental details.

Answer:

We thank the reviewer for pointing out this omission. We agree that a detailed description of the experimental procedure is crucial for reproducibility.

In the revised manuscript, we have supplemented the detailed experimental protocols for the chemical vapor deposition (CVD) modification in the Supporting Information (specifically in Supplementary Experimental Section).

The additional details include specific reaction temperatures and durations, as well as the types and quantities of the silane coupling agents employed, as follows:

Chemical vapor deposition

Two 5 mL centrifuge tubes, each containing 1 mL of methyltrimethoxysilane (MTMS) and 0.8 mL of deionized water, were placed alongside the aerogel samples inside a 1 L desiccator. The desiccator was then hermetically sealed and maintained at 70 °C for 6 h.

Reference

- 1 Du, Y. et al. Sub-1 nm Nanowire Aerogels. *Adv. Funct. Mater.* **35**, 2413651, (2025).

Reviewer: 2

Remarks to the Author

This manuscript reports an intriguing study that employs ultrafine nanowires to construct aerogels, thereby achieving further miniaturization of the aerogel building blocks. The noteworthy results include the successful fabrication of sub-1 nm nanowire-based aerogels with ultralow density, high specific surface area, excellent optical transparency, and remarkable compressive resilience. This work is significant to the fields of porous materials and lightweight functional materials, providing a promising strategy for the rational design of high-performance aerogels and advancing beyond conventional nanoparticle or cluster-based aerogels. Overall, this work is of high quality, and I suggest acceptance after minor revisions addressing the following comments.

1. The preparation method of the SNWs was not introduced either in the main text or in the SI. In particular, the authors should clearly state what specific SNWs are investigated in this work (e.g., Gd₂O₃?). This information should be explicitly provided in the manuscript and in the corresponding figure captions.

Answer:

We sincerely apologize for the omission and thank the reviewer for this constructive suggestion. In the revised manuscript, we have addressed these points as follows:

- (1) **Synthesis Method:** The detailed synthesis procedures for the SNWs have been added to the **Experimental Section** of the Supporting Information.
- (2) **SNW Identification:** We have clarified that the SNWs used in this work are based on rare-earth elements, specifically **Pr, Nd, Sm, Eu, Gd, and Tb**. Among these, **GdOOH SNWs serve as the primary research object**.
- (3) **Manuscript Updates:** The identities of the SNWs have been explicitly clarified in the revised main text (highlighted in yellow) to ensure maximum clarity for the readers.

Among them, the synthesis method of SNWs is as follows:

Synthesis of REOOH sub-1 nm nanowires: REOOH (RE=Pr, Nd, Sm, Eu, Gd, Tb) sub-1 nm nanowires (SNWs) were synthesized through a hydrothermal synthesis method. Here, we took the synthesis of GdOOH sub-1 nm nanowires (Gd-SNWs) as an example. 0.8 g GdCl₃•6H₂O was dispersed in 12 mL C₂H₅OH and 1 mL deionized water. Then added into the mixed dispersion of 4 mL oleylamine (OM) and 2 mL oleic acid (OA) with stirring. Next, the reactants were placed in a 50 mL reactor using 170 °C heating for 4 h. After the reaction, the viscous dispersion was dispersed with cyclohexane and precipitated by centrifugation with C₂H₅OH. The supernatant was then poured off and the precipitate was again dispersed using cyclohexane and centrifuged using C₂H₅OH to purify the SNWs. For the synthesis of other rare-earth SNWs, the amounts of precursors were determined based on the molar ratio used for GdCl₃•6H₂O, while maintaining identical solvent volumes across all preparations. All SNWs in this article were prepared according to the method previously reported by Xun Wang and co-workers^{1,2}.

2. Figures 1b and 1c are presented without sufficient discussion or explanation of the observed phenomena. The authors should elaborate on the key findings illustrated by these figures to enhance the reader's understanding.

Answer:

We appreciate the reviewer's insightful suggestion. We agree that a more detailed discussion of the solvent-dependent behavior of SNWs is crucial for understanding the aerogel formation process. In the revised manuscript, we have expanded the discussion regarding the phenomena observed in Figures 1b and 1c.

Specifically, as illustrated in Fig. 1b, the GdOOH SNWs (Gd-SNWs) exhibit excellent dispersibility in cyclohexane, forming a macroscopically transparent dispersion. This is primarily attributed to the abundant oleic acid (OA) ligands anchored on the SNW surfaces, which provide a compatible interface for uniform dispersion within non-polar media. However, upon exposure to polar solvents, the solvophobic nature of these surface ligands triggers rapid inter-wire aggregation, ultimately culminating in the precipitation of the SNWs.

This detailed explanation has been incorporated into **paragraph 5** (highlighted in yellow) of the revised main text.

Fig. 1 (a) Flowsheet for ligand regulation of SNWs and preparation of SNWA. The dispersion of SNWs in (b) cyclohexane and (c) tert-butanol.

3. The authors employ several characterization techniques to demonstrate the regulation of surface ligands on SNWs. However, solvent-related peaks can still appear even if the SNWs were merely immersed in those solvents. It is therefore recommended that the authors provide additional spectral evidence, such as characteristic peak shifts, to substantiate the proposed surface ligand regulation.

Answer:

We thank the reviewer for this insightful comment. We understand the concern regarding potential interference from residual solvents. To address this, we have provided further clarification on our rigorous sample purification process and **included additional spectroscopic evidence to confirm successful ligand regulation.**

First, to ensure that the observed signals originate from the ligands rather than physisorbed solvent molecules, all SNW samples underwent a stringent purification protocol involving multiple precipitation-redispersion cycles and prolonged thermal treatment.

Specifically:

- (1) For ligand-exchanged SNWs (in tert-butanol): Acetonitrile was used as an antisolvent to precipitate the SNWs. The solids were subjected to three successive cycles of redispersion and centrifugation, followed by drying in an electric thermostatic drying oven at 90 °C for 6 h to eliminate any residual tert-butanol or acetonitrile.
- (2) For pristine SNWs (in cyclohexane): A similar washing process was conducted using

ethanol as the precipitant for three cycles, followed by the same drying procedure (90 °C, 6 h).

This detailed procedure has now been added to the Supporting Information under the section **Sample pretreatment before testing**.

Secondly, in response to the reviewer's suggestion for more spectroscopic evidence, we have included new FTIR data (**Supplementary Fig. S3a, b**). **As shown in the spectra, a distinct evolution in the peak shape and intensity of the hydroxyl (-OH) stretching vibration band near 3400 cm⁻¹ was observed after ligand regulation.** This change is directly attributed to the influence of the 15-HA ligand, rather than any solvent residue, as the solvents used do not possess this specific hydroxyl signature.

Relevant analysis and discussion have been incorporated into the revised manuscript (Paragraph 7, highlighted in yellow), and the data are provided in the Supporting Information (Fig. S3).

Supplementary Fig. S3 (a) FTIR curves of SNWs before and after ligand regulation. (b) FTIR curves of OA and after 15-HA.

4. The manuscript discusses the photoluminescent properties of the aerogels, yet the excitation wavelength used in the measurements has not been specified. As photoluminescence is highly dependent on the excitation wavelength, this information is essential. Moreover, did the authors do the quantized photoluminescence measurement?

Answer:

We sincerely thank the reviewer for pointing out this critical experimental detail. We apologize for the oversight. The excitation wavelength used for UV irradiation in this study was 254 nm. We have now explicitly included this information in the revised manuscript on page 13 (highlighted in yellow). The photoluminescence (PL) emission spectra of Tb-SNWAs and Eu-SNWAs were recorded under excitation wavelengths of 378 nm and 395 nm, respectively, which are provided in **Supplementary Fig. S12a and b**.

Furthermore, the detailed analysis regarding the PL emission and excitation spectra of the Tb-SNWAs has been incorporated into page 17 of the main text, as follows:

To further elucidate the optical performance of the Tb-SNWAs, steady-state PL emission spectra and PL excitation spectra were recorded at room temperature (**Supplementary Fig. S12a**). The PL emission spectra exhibit the characteristic emission fingerprint of Tb³⁺. Four distinct

emission bands are observed at approximately 490, 545, 585, and 622 nm, which are assigned to the radiative transitions from the 5D_4 excited state to the multiple 7F_J ($J = 6, 5, 4, 3$) ground-state manifolds, respectively. Notably, the hypersensitive transition at 545 nm ($^5D_4 \rightarrow ^7F_5$) dominates the spectrum, accounting for the high-purity green luminescence observed macroscopically. The narrow full width at half maximum (FWHM) of these peaks underscores the excellent monochromaticity of the SNWA. The PL excitation spectra, obtained by monitoring the green emission at 545 nm, displays a series of sharp peaks in the near-UV region (300–400 nm). These features correspond to the intra-configurational 4f–4f transitions of Tb^{3+} , such as $^7F_6 \rightarrow ^5D_{2,3}$ and $^7F_6 \rightarrow ^5L_{10}$. Furthermore, the broad excitation band observed in the deep-UV region (<300 nm) can be attributed to the 4f–5d transitions of Tb^{3+} . This wide excitation range facilitates efficient energy harvesting under standard UV illumination, confirming the potential of Tb-SNWAs for advanced optoelectronic applications.

Supplementary Fig. S12. The PL spectrum of aerogel and its optical image under UV light. The PL excitation spectra and emission spectra of (a) Tb-SNWAs and (b) Eu-SNWAs. (c) SNWs gels and aerogels by mixing Eu-SNWs and Tb-SNWs with mass ratios of 1:0, 7:33, 1:1, 3:7, and 0:1 (from left to right).

5. The authors report modifying the SNW aerogels via CVD using MTMS and water; however, the specific processing parameters are not described. Similarly, key conditions for the supercritical drying process such as the type of supercritical fluid, drying temperature, and pressure are missing. These parameters critically influence the pore structure and density of the final aerogels, and are thus necessary for result interpretation and experimental reproducibility. The complete preparation and drying parameters should be included.

Answer:

We sincerely thank the reviewer for pointing out these critical omissions. We agree that providing precise experimental parameters is essential for the reproducibility of our work. In the revised manuscript, we have fully documented the detailed processing conditions as follows:

- (1) **CVD Modification:** For the surface modification, two 5 mL centrifuge tubes, containing 1 mL of methyltrimethoxysilane (MTMS) and 0.8 mL of deionized (DI) water, respectively, were placed in a desiccator alongside the aerogel samples. The desiccator was subsequently sealed and maintained at 70 °C for 6 h to facilitate the vapor-phase deposition.
- (2) **Supercritical Drying (SCD):** The aerogels were fabricated using a supercritical carbon dioxide (CO₂) drying system according to the following procedure. The extraction vessel was initially preheated to 45 °C. Upon sample loading, the internal pressure was increased to 17 MPa at a controlled rate of 0.1 MPa per 30 s and subsequently maintained for a 5 h isothermal dwell. Following this, the temperature was raised in a stepwise manner: first to 55 °C for 30 min, and then to 65 °C for an additional 30 min. After the temperature rise is completed, CO₂ circulation was initiated and sustained until no liquid effluent was discharged. Finally, the vessel was depressurized to ambient pressure at a rate of 0.1 MPa per 30 s to yield the dried aerogels.

6. On page 7, the phrase “an external standard method” is used but not defined. The authors should clarify what this method refers to and provides appropriate references if applicable.

Answer:

We thank the reviewer for this valuable suggestion. To ensure clarity and technical rigor, we have added a specific operational protocol for the external standard method in the revised Supporting Information.

This method was employed to quantify the oleic acid content by establishing a linear relationship between known concentrations and their corresponding signals.

The detailed preparation of the standard solutions is now documented **in the Standard Solution Preparation section of the Supporting Information**. The corresponding regression analysis is now reported in the section entitled **Calculation of the Oleic Acid Content Displaced by the Ligand in the Supporting Information**.

The preparation of the standard solution is as follows:

Preparation of a mixed standard solution I: A mixed standard solution of oleic acid (7 wt%) was prepared using a binary solvent system of acetonitrile and tert-butanol with a volume ratio of 15:3.2. The resulting solution was subsequently stored in a sealed glass vial for further use.

Preparation of a working solution: The initial stock solution I was step-wise diluted with the acetonitrile/tert-butanol co-solvent to yield a series of oleic acid standard solutions with a mass concentration gradient of 0.1, 0.2, 0.3, 0.35, 0.4, 0.45, 0.5, 0.6, and 0.7 wt%. These solutions were stored in sealed vials and subsequently utilized for the construction of a calibration curve.

7. Certain details have been overlooked, which impede comprehension. For example, the label “15-HEA” in Figure 1a is undefined and not explained in the text or figure caption. The authors should ensure that all figure labels are clearly defined to facilitate accurate interpretation by readers.

Besides, a spelling error is present in the label of the horizontal axis in Figure S11, which should be corrected to “Wavelength”.

Answer:

We sincerely thank the reviewer for the keen observation and for pointing out these oversights. We have corrected the errors and provided corresponding clarifications to ensure clarity and accuracy in the manuscript.

- (1) **Label Definition:** We apologize for the ambiguity caused by the label "15-HEA." It refers to 15-hydroxypentadecanoic acid. To improve nomenclature consistency, we have replaced "15-HEA" with "15-HA" in Figure 1a and throughout the manuscript. Its full name is now explicitly defined in both the revised main text and the caption of Fig. 1.
- (2) **Typo Correction:** The spelling error on the horizontal axis of Supplementary Fig. S11 has been rectified, and the label now correctly reads “**Wavelength**”.

We have conducted a thorough check of all figures and labels in the revised manuscript to prevent similar issues.

8. The final section titled “Discussion” should be renamed “Conclusion.” In addition, the writing throughout the manuscript need to be further polished to improve clarity, consistency, and overall readability.

Answer:

We agree with the reviewer’s suggestion. In the revised manuscript, the heading of the final section has been renamed from “Discussion” to “Conclusion” to more accurately reflect the nature of the summary.

We sincerely apologize for the language issues in the previous version. As suggested, the entire manuscript has been thoroughly revised and polished by professional editors at Editage to ensure clarity, consistency, and overall readability.

An editorial certificate from Editage is attached to the submission as proof of this professional polishing. We hope the revised manuscript now meets the high standards of the journal.

References

1. Zhang, S. et al., Highly Flexible and Stretchable Nanowire Superlattice Fibers Achieved by Spring-Like Structure of Sub-1 nm Nanowires. *Adv. Funct. Mater.* **29**, 1903477, (2019).
2. Hu, S., Liu, H., Wang, P. & Wang, X. Inorganic Nanostructures with Sizes down to 1 nm: A Macromolecule Analogue. *J. Am. Chem. Soc.* **135**, 11115-11124, (2013).

Response Letter to Reviewers

We once again thank the Editor for the opportunity to revise our manuscript. Following the reviewers' suggestions, we have carefully and thoroughly revised the work. We now resubmit the revised manuscript together with our point-by-point responses to the comments.

Response to the reviewer's comments:

We once again thank you for your constructive suggestions, which have greatly helped to improve the quality of our manuscript. The changes in the revised manuscript have been highlighted in yellow.

Comments:

Reviewer #1 (Remarks to the Author):

I have reviewed the revised manuscript and the author's response letter. The author responded carefully to the questions I raised. The manuscript is now clearer and the conclusion is more fully supported. Therefore, I suggest accepting it.

Answer:

We sincerely thank Reviewer for the positive assessment of our work and the recommendation for acceptance. We are pleased that our revisions and responses have addressed the reviewer's concerns and that the manuscript is now clearer. We appreciate the time and effort the reviewer dedicated to improving the quality of our study.

Reviewer #2 (Remarks to the Author):

All my previous comments have been addressed. I would only suggest a minor correction: the horizontal axis in Figures S12a and S12b should be labeled as "Wavelength." In addition, the values of 545 nm and 615 nm indicated in the PLE curves in Figures S12a and S12b appear to correspond to the emission wavelengths rather than the excitation wavelength.

Answer:

We thank the reviewer for the careful reading and the helpful suggestions regarding Figures S12a and S12b. We have revised Figure S12 accordingly. The updated figure and the specific changes are shown below:

Supplementary Fig. S12. The PL spectrum of aerogel and its optical image under UV light. The PL excitation spectra and emission spectra of (a) Tb-SNWAs and (b) Eu-SNWAs.